# Challenges in Lipidomics Biomarker Identification: Avoiding the Pitfalls and Improving Reproducibility

**DOI:** 10.3390/metabo14080461

**Published:** 2024-08-19

**Authors:** Johanna von Gerichten, Kyle Saunders, Melanie J. Bailey, Lee A. Gethings, Anthony Onoja, Nophar Geifman, Matt Spick

**Affiliations:** 1School of Chemistry and Chemical Engineering, Faculty of Engineering and Physical Sciences, University of Surrey, Guildford GU2 7XH, UK; j.vongerichten@surrey.ac.uk (J.v.G.); ks00916@surrey.ac.uk (K.S.); m.bailey@surrey.ac.uk (M.J.B.); 2Waters Corporation, Wilmslow SK9 4AX, UK; lee_gethings@waters.com; 3School of Health Sciences, Faculty of Health and Medical Sciences, University of Surrey, Guildford GU2 7XH, UK; a.onoja@surrey.ac.uk (A.O.); n.geifman@surrey.ac.uk (N.G.)

**Keywords:** lipidomics, separation science, mass spectrometry, bioinformatics, machine learning, retention time

## Abstract

Identification of features with high levels of confidence in liquid chromatography–mass spectrometry (LC–MS) lipidomics research is an essential part of biomarker discovery, but existing software platforms can give inconsistent results, even from identical spectral data. This poses a clear challenge for reproducibility in biomarker identification. In this work, we illustrate the reproducibility gap for two open-access lipidomics platforms, MS DIAL and Lipostar, finding just 14.0% identification agreement when analyzing identical LC–MS spectra using default settings. Whilst the software platforms performed more consistently using fragmentation data, agreement was still only 36.1% for MS^2^ spectra. This highlights the critical importance of validation across positive and negative LC–MS modes, as well as the manual curation of spectra and lipidomics software outputs, in order to reduce identification errors caused by closely related lipids and co-elution issues. This curation process can be supplemented by data-driven outlier detection in assessing spectral outputs, which is demonstrated here using a novel machine learning approach based on support vector machine regression combined with leave-one-out cross-validation. These steps are essential to reduce the frequency of false positive identifications and close the reproducibility gap, including between software platforms, which, for downstream users such as bioinformaticians and clinicians, can be an underappreciated source of biomarker identification errors.

## 1. Introduction

The identification of lipids allows for biological interpretation, as well as the association of specific lipids with cellular processes, signaling pathways, and disease conditions [1,2]. In addition, bioinformatics allows for the integration of lipidomic identifications with other omics datasets, such as genomics, proteomics, and metabolomics, to provide a more comprehensive understanding of cellular processes and interactions [3,4]. Therefore, accurate and reproducible identification is critical when searching for biomarkers, features that can indicate the presence or prognostics of a disease, allowing for early diagnosis and personalized medicine. Conversely, inaccurate identification can lead to incorrect conclusions and potentially misleading research findings [5,6,7].

The desire for accurate identification—as in other areas of omics research—has driven the creation of the Lipidomics Standards Initiative (LSI) [8]. This initiative sets out recommended procedures for quality controls, reporting checklists, and minimum reported information [9,10]. However, the LSI is less mature in its recommendations and implementation than, for example, the Metabolomics Standards Initiative (MSI), which itself is still evolving [11,12]. The difficulty in defining parameters for confident annotation partly reflects the sheer range of potential lipids and matrices, encompassing cells, biofluids, tissues, plant extracts, and others [13,14]. This range of samples is then multiplied by the panoply of analytical platforms and separation techniques, such as reversed-phase LC or hydrophobic interaction LC (HILIC). This makes standards-based matching and other best practices more challenging, costly, and time-consuming, especially at the discovery stage.

These issues are well described in the literature, but an underappreciated source of reproducibility problems in untargeted analysis is the lack of consistency in outputs from lipidomics software platforms. Whilst analytical chemists specializing in lipids will often be aware of the issues with peak annotation and feature identification, this will not always be the case for users such as bioinformaticians and clinicians. These software solutions typically pre-process lipid spectra in a five-step workflow previously summarized by Song et al. [15], comprising (i) baseline and noise reduction, (ii) peak identification and extraction, (iii) smoothing, (iv) calculation of signal-to-noise ratios, and (v) isotope identification and deconvolution. Following these steps, accurate mass-to-charge ratio (*m*/*z*) matching is used for identification, combined with fragmentation spectra derived from MS^2^. However, such MS^2^ spectra are not infallible given the potential for co-elution of lipids within the precursor ion selection window and co-fragmentation. Furthermore, MS^2^ may not be practical for lower-abundance lipids. The same abundance issues hinder ion-mobility mass spectrometry [16], a powerful technique that allows for the separation of isobaric lipids but requires specialist instruments and has trade-offs between sensitivity and resolving power. Inconsistencies can also be driven by the use of different lipid libraries such as LipidBlast, LipidMAPS, ALEX123, and METLIN [17]. These issues can also be magnified by different spectral alignment methodologies, which are often opaque to the end user and can cause substantial differences in peak identification. One inter-laboratory comparison of lipidomics LC–MS alignment found an agreement for post-processed features of around 40% between the two laboratories [18].

Another reason for inconsistencies between platforms in untargeted analyses and the need for outlier detection and manual curation is that the majority of lipidomics software tools do not make full use of retention time (*t*_R_), a rich source of information that has been used extensively in machine learning approaches to improve proteomic identifications [19,20,21,22]. Machine learning methods that use algorithms trained on specific columns and operating conditions do not generalize well and are not straightforward to implement across the full range of lipidomics modalities. Whilst the producers of lipidomics software are, of course, aware of these limitations and recommend that putative lipid identifications be manually curated, this can be time-consuming, as well as imposing barriers to entry around lipidomics research for bioinformaticians and clinicians. This is especially the case in secondary analyses such as obtaining validated biomarkers, meta-analyses, or systematic reviews [23,24].

Whilst these issues are recognized by the LSI and other best practice guidelines, there is a paucity of benchmarking of software applications relative to LC–MS methodologies. In this work, we provide a case study in order to reveal the problems researchers face in the form of potentially inaccurate biomarker identifications by processing an identical set of LC–MS spectra using two popular lipidomics platforms: MS DIAL and Lipostar. Whilst workflows for the individual platforms are well-described, to our knowledge, this is the first cross-platform comparison on a single LC–MS dataset. We use this case study to highlight the importance of manual curation and the pitfalls of relying too heavily on ‘top hit’ software identifications, including for discovery work or where groups do not have in-house lipid curation libraries. This can be particularly relevant to new researchers in the field, who may face challenges in receiving sufficient support for analytical training and education [25]. We also demonstrate a novel data-driven quality control step for outlier detection applicable to any untargeted lipidomics analysis by using support vector machine regression combined with leave-one-out cross-validation to identify potentially false positive identifications. These types of quality control steps can be performed on computers typically available in a laboratory setting, without recourse to high-performance computing clusters, and can support manual inspection processes. Such additional steps, especially manual curation, are necessary even where MS^2^ spectra are used, particularly in instances of conflicting identifications by differing platforms.

## 2. Materials and Methods

### 2.1. PANC-1 Lipid Extraction LC–MS Dataset

The lipidomics case study dataset used in this work analyzed a lipid extraction of a human pancreatic adenocarcinoma cell line (PANC-1, Merck, Gillingham, UK, cat no. 87092802). Lipids were extracted by a modified Folch extraction using a chilled solution of methanol/chloroform (1:2 *v*/*v*) according to the protocol described by Zhang et al. supplemented with 0.01% butylated hydroxytoluene (BHT) to prevent lipid oxidation [26]. An Avanti EquiSPLASH^®^ LIPIDOMIX^®^ quantitative mass spectrometry internal standard, a mixture of deuterated lipids, was added to the extract, and the resulting mixture was then diluted to 280 cells/µL. The final EquiSPLASH concentration was 16 ng/mL. Injections of 5 µL of the lipid extract were analyzed using an Acquity M-Class UPLC system (Waters, Wilmslow, UK) coupled to a ZenoToF 7600 mass spectrometer (Sciex, Macclesfield, UK) operated in positive mode. A Luna Omega 3 µm polar C18 column was used (50 × 0.3 mm, 100 Å, Phenomenex, Macclesfield, UK, cat no. 00B-4760-AC) for microflow separation at 8 µL/min. A binary gradient was carried out using eluent A (60:40 acetonitrile/water) and B (85:10:5 isopropanol/water/acetonitrile), both supplemented with 10 mM ammonium formate and 0.1% formic acid. Separation was achieved using the following gradient: 0–0.5 min, 40% B; 0.5–5 min, 99% B; 5–10 min, 99% B; 10–12.5 min, 40% B; 12.5–15 min, 40% B. Mass spectrometry settings are set out in Appendix A. The analysis was conducted in 2023; the untargeted approach described here, including the use of positive mode, was adapted from a commonly used method [27].

The output files were then processed in two lipidomics applications, MS DIAL (v4.9.221218) and Lipostar (v2.1.4) [28,29], using settings set out in full in Appendix A; settings were chosen to make the assumption sets used by the two platforms as similar as possible, but the default libraries were used. For data-driven outlier analysis, a .csv file was prepared for each output, containing the chemical formula for the parent molecule, the class of lipid, the lipid *t*_R_, MS^1^ and MS^2^ status, and the putative identification. Lipids with *t*_R_ below 1 min were considered to have no column retention at all (i.e., eluting with the solvent front) and were excluded from the outlier analysis as having no useful dependent variable.

### 2.2. Comparison of Outputs

Both Lipostar and MS DIAL produced a list of putative identifications based on both MS^1^ and MS^2^ data. All lipidomics (and omics software in general) rely on user settings but also built-in analytical steps for alignment and lipid library access, which may produce inconsistencies. The two output datasets from identical input spectral files were compared to identify the overlapping and unique lipid annotations. Lipid identifications were only considered to be in agreement if the formula was identical, the lipid class was identical, and the aligned retention time was consistent within 5 s between MS DIAL and Lipostar.

### 2.3. Post-Software Quality Control Checks of Data

Post-software quality control steps were then conducted on the assumption that the initial output from the lipidomics software would not represent a ‘definitive’ ground truth. This step—aiming to provide a method and platform-neutral means of improving confidence in lipid annotations—employed a support vector machine (SVM) regression algorithm using leave-one-out cross-validation (LOOCV) in order to predict lipid *t*_R_ [30,31]. The independent variable inputs for the algorithm were the atom count of the parent lipid (i.e., numbers of carbon, hydrogen, nitrogen, oxygen, or other atoms) and lipid class, including inter alia diglycerides (DGs), triglycerides (TGs), phosphatidylcholines (PCs), and ceramides. *t*_R_ was the dependent variable. SVM was chosen for its stability of outputs, ability to deal with multicollinearity, e.g., between carbon and hydrogen atom count (C and H count hereafter), and efficient execution time relative to tree-based algorithms. In addition, given an a priori assumption that the latent variables were linear, using a linear kernel can be preferable to step-function models, which can be more prone to overfitting [32]. Numeric variables were auto-scaled, and categorical variables (lipid classes) were one-hot encoded prior to their inclusion [33]. A linear kernel was used with C = 10, and feature importance was assessed by measuring the explanatory contribution of each variable by SHAP values (SHapley Additive exPlanations), which quantify how much each feature contributes to a model’s prediction of the dependent variable, in this case, *t*_R_ [34,35]. In some cases, SHAP values for a feature can be misleading due to non-confounding redundancy, where a feature explains *t*_R_ but also causally drives another feature included in the model, which in turn also explains *t*_R_ [36].

The code was developed in Python using the scikit-learn (v1.3), shap (v0.43.0), and chemparse (v0.1.2) libraries [37,38,39]. The tqdm (vv4.66.1) library was used to include progress bars for the more time-intensive processes [40]. This code is provided in full as a Jupyter Notebook together with the original raw spectral files and the processed outputs from Lipostar and MS DIAL (as described under Data Availability) and requires no additional software other than a Python environment. The code described in this work was run on a standard Windows PC with a 12th Gen Intel Core i7 CPU paired with 32.0 GB of memory, without employing GPU resources, using the Spyder IDE [41].

As a final step in assessing differences in MS^2^ spectra, lipid identifications were then reviewed based on confidence criteria reported by the two software platforms and then manually inspected in SCIEX (v3.0.0.3339), with a particular focus on (i) outliers identified by the SVM with LOOCV algorithm described above and (ii) lipids where MS DIAL and Lipostar provided conflicting identifications in spite of MS^2^ fragmentation data being available.

## 3. Results

### 3.1. Comparison of MS DIAL and Lipostar Outputs

MS DIAL produced 907 putative lipid identifications across 64 lipid classes from the PANC-1 LC–MS spectra, the most common of which being ceramides (232 identifications across six subclasses), ether PCs (75), and PCs (72). Retention times varied from 1.1 min to 12.4 min, and *m*/*z* values ranged from 153.1 to 898.8. Lipostar produced 979 putative lipid identifications across 43 lipid classes, the most common of which were PCs (151 identifications), DGs (130), and ceramides (114). Retention times for the Lipostar-identified peaks varied from 1.1 min to 12.3 min, and *m*/*z* values ranged from 177.1 to 889.7.

As a simple measure of agreement and disagreement, Figure 1 shows the common (same formula, *t*_R_, and lipid class) and unique identifications for the MS^1^-only features and also the features with MS^2^ data. In total, the two platforms generated 1653 unique identifications, of which 231 were common to both platforms, or 14.0%. The breakdown of the lipid classes by matched and unmatched status is shown in Figure 2.

### 3.2. Data-Driven Investigation of Putative Lipid Identifications

In line with best practice, and given the low numbers of common identifications for the PANC-1 dataset between the two platforms, additional investigations were undertaken. First, both platforms provide a variety of ‘scores’ to help assess confidence in the annotation. For Lipostar, the overall identification score is based on a weighted average of mass score, isotopic pattern score, and fragment score (which itself is a geometric average of the number of fragments score multiplied by ion intensity score) [28]. For MS DIAL, the overall identification score is calculated as a weighted average of MS^2^ similarity, MS^1^ similarity, retention time similarity, and isotopic similarity [42]. The two ‘scores’ are calculated in different ways; therefore, they should not be directly compared, but as shown in Figure 3, there was no clean ‘score’ threshold to identify matched/overlapping features versus non-matched features. Whilst the software platforms generally reported a higher ‘score’ for features identified by both platforms, the overlap was far from perfect.

Next, a platform-independent data-driven approach was adopted by assessing the overall internal consistency of the elution order of the lipid annotations produced by each software package to identify outliers versus expected *t*_R_ values. This was performed without reference to external data, such as custom libraries. The analysis was performed using SVM regression combined with LOOCV, which was applied only to the data internal to the LC–MS spectra analyzed. For the MS DIAL dataset, the algorithm assessed the identifications as being 80.1% internally consistent (i.e., with 19.9% of identifications being further than 5% of the LC–MS runtime from their predicted values). For the Lipostar dataset, the algorithm assessed the identifications as being 69.1% internally consistent, i.e., the lipid identifications showed less internal consistency than with MS DIAL. As a simple metric for overall identification performance, these percentages were consistent with the two platforms being unable to fully validate each others’ identifications (Figure 1B).

The Python code used for this data-driven approach also generated a number of visualizations to support post-software investigation of annotations. These are illustrated in Figure 4 for the MS DIAL dataset. The outlier algorithm identified three categories of outlier lipids where *t*_R_ was potentially inconsistent with the lipid identification provided by the software. The first category of outliers included cases where lipid identifications were outside the range of the vast majority of lipids in their class, given comparable carbon counts. The process of identifying clear outliers can be seen by a simple visual comparison of actual *t*_R_ versus the SVM-predicted *t*_R_ in Figure 4A,B.

The second category of outliers was unexpected variations in elution time in sequences of double bonds. An example is shown in Table 1, where decreasing saturation (increasing hydrogen count) of TGs was associated with increasing *t*_R_. However, in the case of C_51_H_98_O_6_, this relationship was inconsistent. The third category of outliers is related to head group ordering. PCs are formed of a quaternary charged amine and two fatty acid chains. DGs are formed of glycerol with two fatty acid chains. The charged quaternary amine renders PCs more hydrophilic than DGs, and so on a C18 column, PCs with equivalent fatty acid chains would be expected to elute earlier than DGs, not at the same time; an example is shown in Table 1.

SHAP values were used to assess feature importance for the outlier detection algorithm and are summarized in Figure 4D. SHAP value beeswarm plots show the contribution of each variable (C count, headgroup, etc., on the *y*-axis) to each individual forecast of *t*_R_ for each lipid (the model output on the *x*-axis) and provide more individual detail than a plot of overall feature importances. H count was the most explanatory variable, with higher-than-average numbers of H atoms (red) associated with increased model output (predicted *t*_R_). H count was selected by the model over C count, but it should be noted that this is an example of non-confounding redundancy, as both H and C are causally driven by the same latent variable, in this case, the length of acyl chains. Given the information already available from the H count, in practice, increased C count for a given value of H reduced predicted *t*_R_ slightly. This represents the saturation latent variable (i.e., changes in the CH relationship). The SHAP values also show the impact of the headgroup independently—for example, if all other variables were held equal, a PC headgroup would reduce *t*_R_.

### 3.3. Manual Investigation of Putative Lipid Identifications

Following the data-driven steps described in the preceding section, MS^2^ spectra for the lipids were investigated, paying particular attention to any flagged by the SVM outlier analysis or to those with inconsistent identifications between MS DIAL and Lipostar. A number of lipid identifications were found to have MS^2^ spectra inconsistent with the putative software identifications, falling under two headings.

**Co-elution problems**: where several lipids elute at the same time (i.e., within the precursor ion selection window), a variety of lipid fragments may be present in the MS^2^ spectra. An example of this is shown in Table 2 at *t*_R_ 6.78 and 6.80. MS DIAL identified one DG, one TG, and two ceramides. At the same *t*_R_, Lipostar identified three DGs, one TG, and one ceramide. Manual inspection of the MS^2^ spectra (shown in Appendix A) indicated that the four MS DIAL identifications were correct, and so also were three of the Lipostar identifications, i.e., both platforms missed lipids identified by the other platform. Lipostar additionally generated two identifications that could not be validated by manual inspection.

**Insufficient data**: in some instances, the software may simply make an identification where there are insufficient data for a definitive identification. An example is the confusion of PCs and PEs, closely related lipids based on a glycerol backbone, two fatty acid chains, a phosphate group, and a choline or ethanolamine group, respectively. MS DIAL tended to identify these features as PCs, and Lipostar tended to identify the features as PEs. In the example in Table 2 at *t*_R_ 6.00 min, manual inspection of the MS^2^ spectrum indicated that there was insufficient information to be definitive either way (shown in Appendix A).

## 4. Discussion

The ongoing issues around reproducibility in biostatistics and bioinformatics are well-described [43,44]. These issues have a number of causes, such as insufficient documentation, inappropriate applications of hypothesis testing, or poor study design. These challenges also extend to metabolomic and lipidomic biomarker identifications, with consequences for reproducibility when developing diagnostic and prognostic panels [5,45,46]. The challenge can be further exacerbated by the issue shown in this work of inconsistent identifications being produced by different software platforms, even from the same spectral data. Here, agreement on lipid identifications for a bulk cell lysate between Lipostar and MS DIAL, two open-source lipidomics platforms in common use, was just 14.0% overall. In addition, in-built scores for identification confidence were, in our view, insufficient for the complexities of the issue at hand, warranting further steps to define the data. The headline differences are partly attributable to different underlying databases (MS DIAL partly uses LipidBlast [47], and LipoStar uses LipidMAPS [48]). Inconsistencies can be reduced through the exclusion of MS^1^ identifications, careful manual curation, and experimental iteration. Nonetheless, the lack of consistency can still present a challenge to researchers.

One approach to dealing with potential problems in LC–MS analysis is outlier detection. The novel SVM regression with the LOOCV method described here successfully identified the major physicochemical properties governing elution order. This was achieved by using H count for acyl length, the CH relationship to identify saturation as a latent variable [49], and automatically identifying the hydrophobicity of lipid headgroups and their influence on *t*_R_, for example, correctly ordering PC and DG headgroups [50]. Crucially, the algorithm can incorporate all these latent variables in its decision-making instead of relying on one variable for assessment; a comparison based solely on C count, for example, cannot provide significant information about *t*_R_. Interestingly, whilst equivalent carbon number and its relationship with retention time has been proposed as a means of verifying *t*_R_ [51], the data-driven approach described here finds better performance from H count in identifying the latent variable (in this case, acyl length), with C count exhibiting non-confounding redundancy. For MS DIAL, outlier detection estimated the peak identifications as 19.9% internally inconsistent and the Lipostar peak identifications as 30.9% internally inconsistent. Taken in combination with the limited overlap between identifications of just 14%, these observations are strongly suggestive that a significant proportion of ‘top hit’ lipid identifications would pose reproducibility problems.

MS^1^ spectra are generally deemed insufficient for lipid identification, especially for QTOF instruments, which in many cases have lower mass accuracy than Orbitrap mass spectrometers. Consequently, the best practice in lipidomics is to use only MS^2^ identifications [6], but even here, the agreement was not perfect. Only 36.1% of MS^2^ identifications could be matched between MS DIAL and LipoStar, and the outlier detection algorithm found that 5.3% of MS DIAL and 22.4% of Lipostar putative MS^2^ identifications were internally inconsistent. This lack of consistency in MS^2^ identifications—both internally and between platforms—was driven by co-elution issues and closely related lipids being difficult to distinguish from each other. Consistency was greatest for the major lipid classes, especially TGs and PCs, and lowest for minor lipid classes. This inconsistency between different software platforms for MS^2^ identifications presents a clear challenge for the interpretation of results, especially for bioinformaticians and clinicians who are less familiar with LC–MS workflows. These results demonstrate that ‘top hit’ identifications by a single software platform should never be taken as a ‘given’ by users of lipidomics research. Based on these findings, software versions and settings are as critical to best practice in reporting as instrumental settings and yet are not given the same prominence in lipid research or in guidelines such as the Lipidomics Minimal Reporting Checklist [10].

Many existing strategies exist to deal with the problem of misidentifications. Visualization tools such as Kendrick mass defect plots can help spot outliers but do not take into account the full range of variables available to the statistical learning approach described here [52]. Other strategies include the use of internal standards, but such standards are expensive, often only offer a small number of standards per lipid class, and can be deployed in a more focused manner once appropriate targets have been identified at the discovery stage. In this case study, the Avanti EquiSPLASH^®^ LIPIDOMIX^®^ standard was used, but this includes only 14 deuterated lipids in the major lipid classes, for example, including just one DG. This can confirm overall LC–MS performance and identify the rough *t*_R_ range for a lipid class within a run but is of lesser use in identifying specific lipids—increasing the number of deuterated standards can be prohibitively expensive, especially in discovery work. Inevitably, inaccurate identifications at the discovery stage pose costs and challenges later on. Data-driven approaches either rely on multiple repeats of the experiment to provide separate training and test sets such as QSSR [53] or the use of specific libraries for samples or methods, for example, data on *t*_R_ values for human plasma lipids or on collision cross sections [54,55]. These library-driven approaches will not reflect the wide range of analytical columns, mobile phases, platforms, and sampling matrices, especially at the discovery stage.

These results also demonstrate that reporting software platform confidence ‘scores’ can be helpful but is insufficient for definitive identification (and, at worst, may produce unwarranted confidence). In addition to these automated steps of outlier detection and reporting of confidence criteria, manual inspection is essential for all potential biomarkers of interest. For example, a manual inspection can check for biological consistency, an important step in lipid review, e.g., flagging the presence of plant-based lipids such as sulfolipids as incongruous in a human plasma sample [56]. It will also frequently be the only means of checking the validity of outlier observations, distinguishing closely related lipids such as PCs and PEs, and resolving problems with co-eluting lipid identifications. There may even be merit in analyzing data using more than one lipidomics software platform—as shown here, neither MS DIAL nor Lipostar alone showed the level of sensitivity or specificity for lipid identifications that would be required for reproducible identifications. Other solutions include extending chromatography run-time, which can help with the co-elution of lipids; running samples in negative mode, as well as positive mode, is also strongly recommended for improved feature annotation. While not an option for the ZenoToF 7600 instrument used in this case study, for some instruments, polarity switching can also be used to offer positive and negative modes within a single run [57]. As with increasing the number of deuterated standards, these solutions can involve cost and sensitivity trade-offs and often require different gradients or phases. As an example, ammonium acetate is better for negative mode, whereas ammonium formate was used here. For targeted work, naturally, such steps become more practical.

This case study only compares two lipidomics platforms, MS DIAL and Lipostar. Many other platforms are available, such as Progenesis QI or LipidSearch, and a more comprehensive exercise to benchmark the full range of platforms across multiple test datasets (covering different biological matrices and instrument methodologies) would have considerable value in highlighting the strengths and weaknesses of each. As previously noted, operating in negative mode as well as positive mode would improve consistency, as certain lipid classes, such as TGs, ionize well only in positive polarity, whilst negative polarity can produce better outputs for phospholipids. In addition, both MS DIAL and Lipostar can import different libraries, and harmonizing the libraries used would reduce differential identifications. These are all issues that could be further addressed in future work and investigations. Nonetheless, our emphasis here is to highlight that in untargeted lipidomics—especially when using default settings and libraries—the complexity of multiple theoretical lipid identifications is one of the major pitfalls for (new) investigators. Many lipid MS studies contain serious errors [58], and this emphasizes the importance of the additional curation steps outlined here.

## 5. Conclusions

In this work, we demonstrate the challenges for reproducibility derived from the choice of lipidomics software platform, an under-investigated source of inconsistencies when identifying lipid biomarkers of interest. This is an important issue, especially for bioinformaticians and clinicians (or indeed generalist readers) when using analytical LC–MS outputs. We also show that a data-driven workflow for outlier detection can learn the latent variables that govern the order of elution and *t*_R_, but manual curation will still be required. This is especially the case where MS^2^ data are challenged by co-elution issues or where lipid classes are similar. In-built software scoring and checks are helpful but, in our view, insufficient, necessitating additional quality control workflow steps. These are essential to reduce inconsistencies in identifications when different groups use different lipidomics platforms, to address problems with reproducibility and replicability for end users of LC–MS data, and to improve confidence in bioinformatics analyses using lipid biomarkers.

## Figures and Tables

**Figure 1 metabolites-14-00461-f001:**
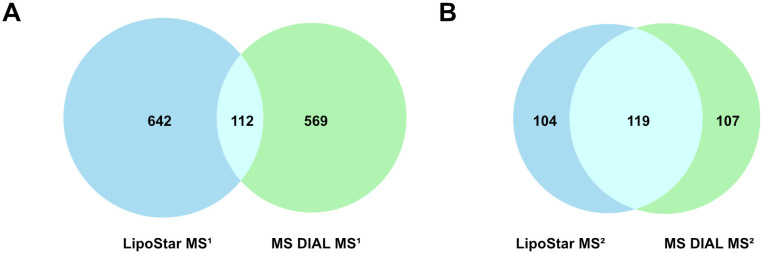
Distinct and overlapping identifications between Lipostar and MS DIAL. (**A**) MS^1^ data only and (**B**) MS^2^ data only.

**Figure 2 metabolites-14-00461-f002:**
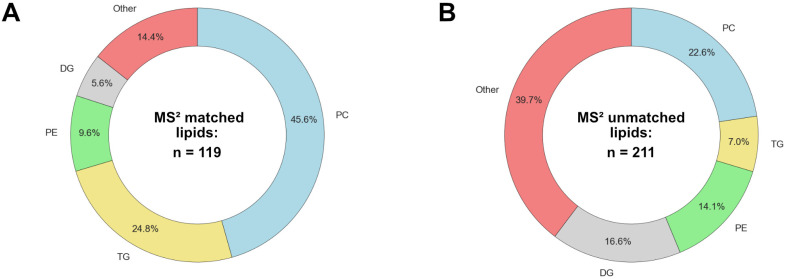
Breakdown of lipid classes identified: (**A**) common MS^2^ identifications and (**B**) unique MS^2^ identifications.

**Figure 3 metabolites-14-00461-f003:**
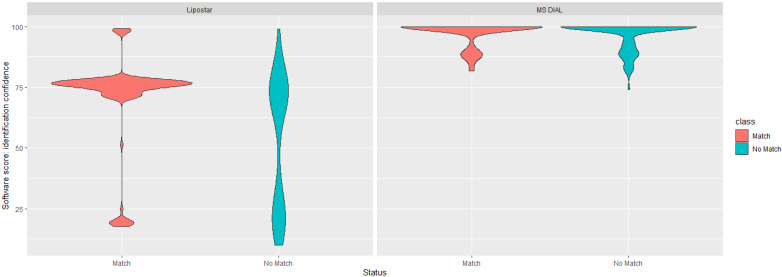
Violin plots for annotation confidence scores of matching and non-matching lipid identifications for Lipostar and MS DIAL. Individual scores range from 0 to 100. MS^2^ identifications are only shown here.

**Figure 4 metabolites-14-00461-f004:**
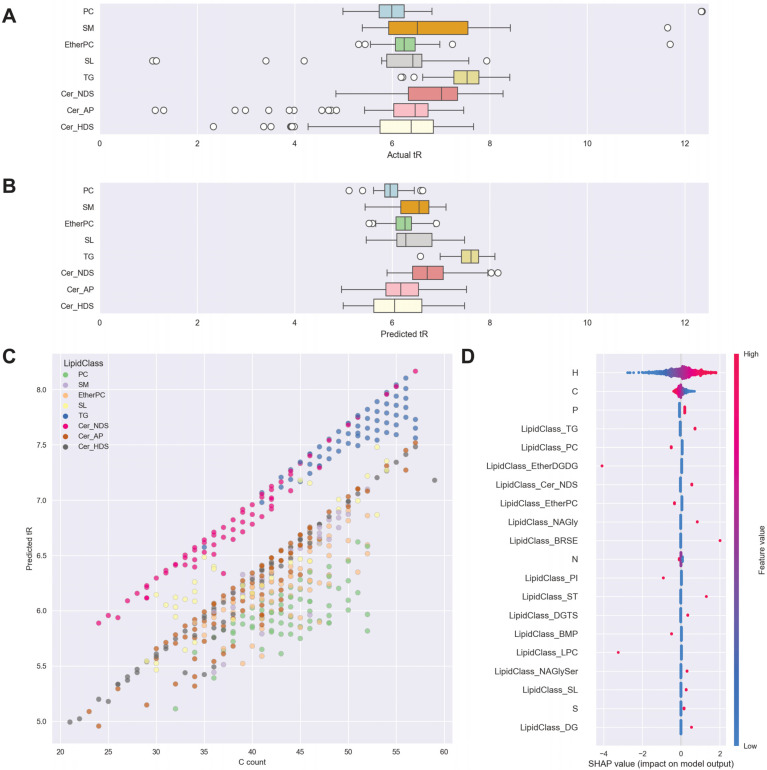
(**A**,**B**) Actual and predicted *t*_R_ plotted as boxplots by lipid class: 8 most abundant lipid classes shown. The upper and lower bounds of boxes show the interquartile range. (**C**) predicted *t*_R_ plotted against carbon atom count, with lipid class shown by color, for 8 most abundant lipid classes. (**D**) SHAP value beeswarm plots for feature importance, atom count, and 8 most abundant lipid classes shown—*x*-axis represents the impact on model output (predicted *t*_R_), each dot represents a sample. The color scale indicates the impact of feature value, where red is an above-average value for a feature such as carbon count, and blue is a below-average value for a feature. TG—triglycerides, SM—sphingomyelins, SL—sphingolipids, PC—phosphatidylcholines, Cer—ceramides including alpha-hydroxy ceramides (Cer_AP), non-hydroxy ceramides containing dihydrosphingosine (Cer_NDS), and hydroxy ceramides containing dihydrosphingosine (Cer_HDS).

**Table 1 metabolites-14-00461-t001:** Example lipids flagged for review versus those with an internally consistent *t*_R_—MS DIAL dataset.

MS DIAL Identified Features	Actual *t*_R_ (min)	Predicted *t*_R_ (min)	Δ (min)
Inconsistencies in *t*_R_: saturation			
TG C_51_H_92_O_6_	7.42	7.49	0.07
TG C_51_H_94_O_6_	7.55	7.57	0.02
TG C_51_H_96_O_6_	7.69	7.65	−0.04
**TG C_51_H_98_O_6_**	**6.62**	**7.74**	**1.12**
Inconsistencies in *t*_R_: headgroups			
DG C_36_H_61_D_7_O_5_	6.55	6.52	−0.03
**PC C_36_H_64_NO_8_P**	**6.50**	**5.42**	**−1.08**

Bold text indicates >5% run-time Δ, flagged for review.

**Table 2 metabolites-14-00461-t002:** Conflicting identifications: MS DIAL versus Lipostar.

MS DIAL Identified Features	*t*_R_ (min)	Lipostar Identified Features	*t*_R_ (min)
Identification problems: co-elution of lipids			
DG 34:0|DG 16:0_18:0	6.78	DG (15:0/16:0/0:0)	6.78
TG 41:1;O|TG 9:0_17:0_15:1;O	6.78	DG (15:1/18:1/0:0)	6.78
Cer 42:2;O2|Cer 18:1;O2/24:1	6.80	TG (13:0/13:0/16:0)	6.78
Cer 42:2;O2|Cer 18:1;O2/24:1	6.80	DG (15:0/18:1/0:0)	6.79
		Cer (51:1)	6.80
Identification problems: misidentifications			
PC 37:7|PC 15:1_22:6	6.00	PE (40:7)	6.00

## Data Availability

The Python code in .ipynb notebook format, as well as the raw mass spectrometry files, are available at the Zotero repository with the following URL: https://zenodo.org/records/10889321 (accessed on 28 March 2024); DOI: 10.5281/zenodo.10889320.

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
