# Peer review of "Challenges in Lipidomics Biomarker Identification: Avoiding the Pitfalls and Improving Reproducibility"

_metabolites, 2024, doi:10.3390/metabo14080461_

Round 1

Reviewer 1 Report

Comments and Suggestions for Authors

The manuscript submitted by von Gerichten et al., nicely demonstrates the challenges and pitfalls of lipid annotation/identification in lipidomics workflows. In this work, the authors utilize two open-source platforms, MS-DIAL and Lipostar, to aid in lipid identification using identical data acquired from Panc1 cells on a RP lipidomics LC-MS method. To compare the list of putative identifications resulting from both platforms, the authors used a series of criteria to determine common vs. unique lipid annotations – to deem the annotations common or identical among both platforms, the putative identifications must have i) an identical chemical formula, ii) an identical lipid class and iii) an aligned retention time that was consistent. Furthermore, post-software quality control checks were implemented using machine learning methods. Based on these criteria, the authors observed that only 14% of the lipid annotations agreed between both platforms – which is a striking result. The authors attribute these inconsistencies in lipid identification due to differences in databases used, co-elution problems and insufficient data. Overall, this manuscript is one of the first papers (to my knowledge) that explores the inconsistencies between different lipid annotation platforms and thus, serves as a meaningful contribution to the lipidomics community.

A few minor comments on the manuscript:

1.     In the Materials and Methods section of the manuscript, was the data acquired using one polarity or both polarities? If acquired on both positive and negative polarity, was polarity switching used or were they acquired separately? Furthermore, if two polarities were used, how were duplicates filtered out (e.g. specific lipid species that were detected on both positive and negative mode)?

2.     I’m assuming DDA MS2 was used to aid in lipid identifications. If so, was iterative MS2 used and if so, how many injections were performed?

3.     In Section 3.2 of the manuscript, could you specify what the overall internal consistency means? Could you provide a specific example to illustrate “acceptable” internal consistency?

4.     Could the authors explain the SHAP plots in Figure 4D in a bit more detail? I’m not familiar with these plots and am still unsure of how this provides insight into the overall figure. Any time of example would be useful.

5.     Please fix the formatting on Section 3.3 - Manual investigation of putative lipid identifications to match the rest of the manuscript

6.     The authors mentioned that MS-DIAL and LipoStar use different lipid databases. Have you tried uploading the same library to both platforms and seeing if this helps increase the coverage of common lipid identifications among both platforms? For instance, did you upload the LipidMAPS library to MS-DIAL (if possible) or vice-versa? I know the LipidBlast library can be freely downloaded as an .msp or .sdf format from the MoNA website and can be uploaded into mzMine, NIST etc. I’m not sure if this is possible on LipidStar.

Aside from these minor comments, this paper was extremely insightful and expect this manuscript to receive positive feedback from the lipidomics community.

Author Response

Challenges in lipidomics biomarker identification: avoiding the pitfalls and improving reproducibility

We would like to express our thanks to the reviewers for taking the time to review our manuscript. Please find the detailed responses below; the corresponding revisions are included with track changes enabled in the re-submitted files (both the manuscript and the supplementary information)

REVIEWER 1

The manuscript submitted by von Gerichten et al., nicely demonstrates the challenges and pitfalls of lipid annotation/identification in lipidomics workflows. In this work, the authors utilize two open-source platforms, MS-DIAL and Lipostar, to aid in lipid identification using identical data acquired from Panc1 cells on a RP lipidomics LC-MS method. To compare the list of putative identifications resulting from both platforms, the authors used a series of criteria to determine common vs. unique lipid annotations – to deem the annotations common or identical among both platforms, the putative identifications must have i) an identical chemical formula, ii) an identical lipid class and iii) an aligned retention time that was consistent. Furthermore, post-software quality control checks were implemented using machine learning methods. Based on these criteria, the authors observed that only 14% of the lipid annotations agreed between both platforms – which is a striking result. The authors attribute these inconsistencies in lipid identification due to differences in databases used, co-elution problems and insufficient data. Overall, this manuscript is one of the first papers (to my knowledge) that explores the inconsistencies between different lipid annotation platforms and thus, serves as a meaningful contribution to the lipidomics community.

A few minor comments on the manuscript:

  1.     In the Materials and Methods section of the manuscript, was the data acquired using one polarity or both polarities? If acquired on both positive and negative polarity, was polarity switching used or were they acquired separately? Furthermore, if two polarities were used, how were duplicates filtered out (e.g. specific lipid species that were detected on both positive and negative mode)?

Positive mode only was used, which is a limitation. Also in response to the other reviewers, we have both added that positive mode was used, and updated supplementary material on methods. Operating in both modes would allow for additional manual curation, and we have revised the limitations section to make this clear.

  1.     I’m assuming DDA MS2 was used to aid in lipid identifications. If so, was iterative MS2 used and if so, how many injections were performed?

Summarising from Sciex’s instrument notes, the Zeno trap is not used iteratively during TOF MS scans because the ion current is already attenuated via ion transmission control (ITC) and the likelihood of detector saturation is high. Once the ions are trapped (t < 1 ms), they are sequentially scanned from the trap to the TOF accelerator in a mass-dependent manner. High molecular weight ions are scanned out first and are followed by lower molecular weight ions, such that they all arrive at the TOF accelerator at the same time. Once the ions are accelerated into the TOF, the Zeno trap fills and repeats the cycle. This process improves the instrument duty cycle and ~90-95% of all fragment ions hit the detector.

We have added a link to Sciex’s technical notes in supplementary material, as we think the explanation is probably too detailed / unwieldy for the manuscript, but it is indeed an interesting point regarding how different instruments approach the problem in different ways.

https://sciex.com/tech-notes/life-science-research/metabolomics/untargeted-data-dependent-acquisition--dda--metabolomics-analysi

  1.     In Section 3.2 of the manuscript, could you specify what the overall internal consistency means? Could you provide a specific example to illustrate “acceptable” internal consistency?

We have added the following text to Section 3.2.

“Next, a platform-independent data driven approach was adopted, by assessing the overall internal consistency of elution order of the lipid annotations produced byof each software output for outlierspackage, to identify outliers versus expected tR values. This was done without reference to external data, such as custom libraries.”

  1.     Could the authors explain the SHAP plots in Figure 4D in a bit more detail? I’m not familiar with these plots and am still unsure of how this provides insight into the overall figure. Any time of example would be useful.

We have added the following text underneath Figure 4D

“SHAP values were used to assess feature importance for the outlier detection algorithm, and are summarised in Figure 4D. SHAP value beeswarm plots show the contribution of each variable (C count, headgroup etc on the y-axis) to each forecast of tR for each individual lipid (the model output on the x-axis) and provide more individual detail than a plot of overall feature importances.”

  1.     Please fix the formatting on Section 3.3 - Manual investigation of putative lipid identifications to match the rest of the manuscript

We have amended the formatting.

  1.     The authors mentioned that MS-DIAL and LipoStar use different lipid databases. Have you tried uploading the same library to both platforms and seeing if this helps increase the coverage of common lipid identifications among both platforms? For instance, did you upload the LipidMAPS library to MS-DIAL (if possible) or vice-versa? I know the LipidBlast library can be freely downloaded as an .msp or .sdf format from the MoNA website and can be uploaded into mzMine, NIST etc. I’m not sure if this is possible on LipidStar.

We have not done this, and agree that it would be interesting to isolate the differences due to libraries and the differences due to software package. We made a conscious decision to use the libraries in their “out of the box” state, i.e. as they would be downloaded for a fresh install of Lipostar and MSDIAL, to illustrate the problems that those new to lipidomics research might encounter. We also worried that if we harmonised the libraries, there would be no guarantee that we knew which library was the ‘most correct’ one! It would be interesting to investigate the issue. We hope to use this work as a pilot study of the problems, and in the future compare a wider range of platforms. Testing the platform versus library difference at this point would be very worthwhile.

Aside from these minor comments, this paper was extremely insightful and expect this manuscript to receive positive feedback from the lipidomics community.

We are very appreciative of the positive comments.

REVIEWER 2

The systematic review about Challenges in lipidomics biomarker identification is an interesting work and the authors have collected unique information from various electronic sources. The review is generally well written and structured. However, in my opinion it has some shortcomings in regards to some data collections and text, and I feel the topic has not been utilized to its full extent. Below I have provided numerous remarks on the manuscript which should be addressed-

  1.       As with all MS you will need to clearly spell out the specific ‘Challenges in lipidomics biomarker identification’ relevance of the topic. Why is this review needed? Is the review related to only targeted biomarker discovery or untargeted too? Clearly state this.

The review is needed in our view because reproducibility of lipid biomarkers has been low (limited translation of discovery work to clinical usage), and because many lipid MS studies are performed incorrectly in general, see for example this reference which we have added to the manuscript.

https://doi.org/10.1038/s41580-024-00758-4

We thank the reviewer for highlighting the point around targeted work, and indeed, this work is only relevant for untargeted / discovery work, as for targeted work authentic standards should have been used to ensure matched retention times. We have made text edits to make clear this work refers to untargeted work.

  1.       The abstract must include the timeline (year) of the source from which information is gathered.

This work did not really have a timeline, it was an experiment conducted in 2023 and all results date from that point. We have added this information to Methods. We hope that this is sufficient to address the reviewer’s concerns.

  1.       The introduction should provide the relevant background and leads to well-defined objectives. Do not report results in the introduction or make conclusions in this section.

We have removed some of the text from the end of the Introduction to address the reviewer’s concerns.

  1.       In general, the language needs some minor revisions. In many parts it is basically unintelligible or one has to guess what the meaning is.

We have made a number of edits to reflect the reviewer’s concerns, also those of reviewer 1 and 3. We hope that this is sufficient. If there are any remaining sentences that are problematic, we will certainly review them.

  1.       Overall we expect a critical assessment of the state of the art including precise and critical assessment of the papers reviewed (incl. concepts and methods).

We have included a number of additional comments (see also responses to reviewers 1 and 3), and included this current reference to issues in lipid MS work, published recently:

https://doi.org/10.1038/s41580-024-00758-4

  1.       Page 10, line 341, support your statement with latest reference- https://doi.org/10.1016/j.ebiom.2023.104627

We have included the requested reference on validated biomarkers “A pentasaccharide for monitoring pharmacodynamic response to gene therapy in GM1 gangliosidosis” but felt that this was more appropriately referenced in the Introduction, and so have included it on page 2, we hope that this meets the reviewer’s concern.

  1.       Conclusions need to be critical and specific. It needs to highlight the achievements and specific scientific gaps in our knowledge. So what further research should have priority?

We thank the reviewer for highlighting this, and have extended the ‘future work’ text extensively:

“It should be noted that this case study only compares two lipidomics platforms, MS DIAL and Lipostar. Many other platforms are available such as Progenesis QI or LipidSearch, and a more comprehensive exercise to benchmark the full range of platforms across multiple test data sets (covering different biological matrices and instrument methodologies) would have considerable value in highlighting the strengths and weak-nesses of each. In particular, operating in negative mode as well as positive mode would improve consistency, as certain lipid classes, such as TGs, ionize well only in positive polarity, whilst negative polarity can produce better outputs for phospholipids. In addition, both MS DIAL and Lipostar can import different libraries, and harmonizing the libraries used would reduce differential identifications. These are all issues that could be further addressed in future work and investigations.”

REVIEWER 3

The article is interesting, but some modifications and further clarifications are needed before its publication in Metabolites. I find the approach convincing, and the development of software or tools that provide information using orthogonal approaches to mass spectrometry, such as retention time, is highly appreciated by those working in lipidomics.

However, the most important question concerns the choice of polarity. While certain lipid classes, such as TGs, ionize well only in positive polarity, negative polarity is generally preferred for phospholipids. Indeed, negative polarity provides significantly more informative and less ambiguous MS/MS spectra and allows for regiochemical attributions, as reported in the studies by Hsu and Turk.

We agree with the reviewer. We have amended the text to make it more clear that running in positive and negative mode is an important part of manual curation. We have observed, however, that in discovery studies this is not always done. Occasionally this is due to sample insufficiency, although this is rare and mostly relates to niche cases such as single cell studies, and sometimes perhaps it is due to cost issues as running in both polarities may require double the injections and two columns (some instruments like Orbitraps offer switching in a single run, but this has trade-offs). Our work here is to illustrate the pitfalls caused by lipid annotation software – it is certainly true that by running in negative mode and manually curating the identifications, by harmonising the libraries, we could reduce the inconsistencies between the platforms, which any large lipidomics group would certainly do (or use their own libraries).

This work is intended to provide a practical example of how poor the overlap can be when annotation packages are used ‘out of the box’, but we have extended the text including under limitations to make clear that negative mode is an important solution to the problem that we are highlighting.

Therefore, why did the authors choose to work only in positive polarity? I believe the article and the conclusions regarding the comparisons of the MS/MS spectra would be much more convincing if conducted in negative polarity. This approach would likely yield more reproducible results across the two platforms and offer a better comparison of the software capabilities.

Again, we agree. See also our response to Reviewer 1, Question 6. There is an interesting follow-up work which could act as a sort of flowchart of errors, tracking down where misidentifications are due to different libraries, where due to different polarities, etc. This was a small pilot study that we thought was interesting to highlight the issue in a quantitative way, and to share the Python code that we have developed to help with the assessment process. We think this is still helpful, given the pitfalls in lipid MS analyses (e.g. see this paper published only this week: https://doi.org/10.1038/s41580-024-00758-4)

I would therefore ask the authors to enhance the article by doubling the comparison and using negative polarity in the identifications based on the MS/MS spectra, to discuss how (I imagine) the two software tools would yield much greater concordance. I know this modification probably requires new acquisitions and work. However, I want to emphasize that I consider this a necessary addition to significantly strengthen the article, which, as I repeat, has a very interesting foundation.

We don’t have the budget or instrument time to rerun the experiment in negative mode. But we do agree with the point that the reviewer is making.

We hope that by highlighting the issue, and flagging it for future work, this work will be acceptable as a ‘first step’ in analysing the role of software platforms and libraries in lipid annotation inconsistency. To our knowledge, there is no equivalent quantitative analysis in the literature, so we believe this work is still helpful to new researchers.

The authors also report the comparison between the DG present in the Equisplash mixture and a PC with an equivalent total number of C atoms as an attribution flagged as inconsistent based on retention times. While I believe that the reported compound is indeed not a PC, why did the authors compare the retention of a DG 33:1 with that of a hypothetical PC 28:4? Does the software compare retentions based on sum composition (which would make sense) or based on the total number of carbon atoms (as it seems to do)?

The Python code takes account of all factors simultaneously, i.e. carbon / hydrogen count, headgroup, saturation. We have added some extra text to the manuscript to make this more clear. “Crucially, the algorithm can incorporate all these latent variables in its decision-making, instead of relying on one variable (such as C count) for assessment.”

Other minor modifications:

In Table S1, it would be useful to indicate that the analyses were conducted in positive polarity.

We have added this clarification.

Please clarify the final concentration of Equisplash in the final solution.

The final EquiSPLASH concentration was 16 ng/mL. We have added this to Methods.

This has been added to Methods.

In the initial part of the work, there is a reference to the use of GC for lipid characterization; this technique should be removed from the discussion as it is used for fatty acids and not intact lipids. The same applies to more niche applications such as chiral columns or "silver ion-LC," which I am not familiar with.

We have removed this from the text.

The captions of the supplementary figures should be enriched with more information. Specifically, in Figure S2, the MS/MS spectrum seems to be consistent with that of a PE 40:7, and the ion at m/z 184 suggests the partial coelution of an isobaric PC. This information could also be added to the text.

We have amended the caption as requested.

Reviewer 2 Report

Comments and Suggestions for Authors

The systematic review about Challenges in lipidomics biomarker identification is an interesting work and the authors have collected unique information from various electronic sources. The review is generally well written and structured. However, in my opinion it has some shortcomings in regards to some data collections and text, and I feel the topic has not been utilized to its full extent. Below I have provided numerous remarks on the manuscript which should be addressed-

1.       As with all MS you will need to clearly spell out the specific ‘Challenges in lipidomics biomarker identification’ relevance of the topic. Why is this review needed? Is the review related to only targeted biomarker discovery or untargeted too? Clearly state this.

2.       The abstract must include the timeline (year) of the source from which information is gathered.

3.       The introduction should provide the relevant background and leads to well-defined objectives. Do not report results in the introduction or make conclusions in this section.

4.       In general, the language needs some minor revisions. In many parts it is basically unintelligible or one has to guess what the meaning is. 

5.       Overall we expect a critical assessment of the state of the art including precise and critical assessment of the papers reviewed (incl. concepts and methods).

6.       Page 10, line 341, support your statement with latest reference- https://doi.org/10.1016/j.ebiom.2023.104627

7.       Conclusions need to be critical and specific. It needs to highlight the achievements and specific scientific gaps in our knowledge. So what further research should have priority?

Comments on the Quality of English Language

Minor editing in English is required

Author Response

(The authors gave the same response as above.)

Reviewer 3 Report

Comments and Suggestions for Authors

The article is interesting, but some modifications and further clarifications are needed before its publication in Metabolites. I find the approach convincing, and the development of software or tools that provide information using orthogonal approaches to mass spectrometry, such as retention time, is highly appreciated by those working in lipidomics.

However, the most important question concerns the choice of polarity. While certain lipid classes, such as TGs, ionize well only in positive polarity, negative polarity is generally preferred for phospholipids. Indeed, negative polarity provides significantly more informative and less ambiguous MS/MS spectra and allows for regiochemical attributions, as reported in the studies by Hsu and Turk.

Therefore, why did the authors choose to work only in positive polarity? I believe the article and the conclusions regarding the comparisons of the MS/MS spectra would be much more convincing if conducted in negative polarity. This approach would likely yield more reproducible results across the two platforms and offer a better comparison of the software capabilities.

I would therefore ask the authors to enhance the article by doubling the comparison and using negative polarity in the identifications based on the MS/MS spectra, to discuss how (I imagine) the two software tools would yield much greater concordance. I know this modification probably requires new acquisitions and work. However, I want to emphasize that I consider this a necessary addition to significantly strengthen the article, which, as I repeat, has a very interesting foundation.

The authors also report the comparison between the DG present in the Equisplash mixture and a PC with an equivalent total number of C atoms as an attribution flagged as inconsistent based on retention times. While I believe that the reported compound is indeed not a PC, why did the authors compare the retention of a DG 33:1 with that of a hypothetical PC 28:4? Does the software compare retentions based on sum composition (which would make sense) or based on the total number of carbon atoms (as it seems to do)?

Other minor modifications:

  • In Table S1, it would be useful to indicate that the analyses were conducted in positive polarity.
  • Please clarify the final concentration of Equisplash in the final solution.
  • In the initial part of the work, there is a reference to the use of GC for lipid characterization; this technique should be removed from the discussion as it is used for fatty acids and not intact lipids. The same applies to more niche applications such as chiral columns or "silver ion-LC," which I am not familiar with.
  • The captions of the supplementary figures should be enriched with more information. Specifically, in Figure S2, the MS/MS spectrum seems to be consistent with that of a PE 40:7, and the ion at m/z 184 suggests the partial coelution of an isobaric PC. This information could also be added to the text.

Author Response

(The authors gave the same response as above.)

Round 2

Reviewer 3 Report

Comments and Suggestions for Authors

Although I understand that repeating certain analyses may not be feasible, I maintain that a comparison between software should be conducted by analyzing their behavior under optimal conditions. Similarly, the databases used should be the same to ensure that the software is evaluated under comparable conditions.

Furthermore, in response to the objection raised in the previous round, it is important to emphasize that in RPC, the separation of lipid species is primarily determined by the identity of the acyl chains and, to a lesser extent, by the sum composition. Therefore, a comparison based solely on the total number of carbon atoms cannot provide significant information about retention.

For these reasons, as I have previously stated, I do not consider the article suitable for publication.

Author Response

Reviewer: Although I understand that repeating certain analyses may not be feasible, I maintain that a comparison between software should be conducted by analyzing their behavior under optimal conditions. Similarly, the databases used should be the same to ensure that the software is evaluated under comparable conditions.

Reply: Respectfully, we disagree. Our experience from the literature is that many published lipidomics research papers use default settings for software solutions for annotations. It is also quite common for published research to use one modality (RP or HILIC) rather than multiple modalities. Of course, experienced lipidomics groups DO use optimised methods, but experienced groups are not our target audience. In short, we believe that there is value in analysing performance under imperfect conditions and not just choosing optimal conditions. We have added some text to the Introduction to make this more clear, and we hope that this is satisfactory.

As no-one else has published a comparison of possible reproducibility issues from lipid software platforms and libraries, we think this is a novel piece of work that can contribute to the debate. A further analysis breaking down more software platforms and libraries (e.g. adding Compound Discoverer, Progenesis, LipidSearch) would have considerable value and we hope to obtain funding for such a work in the near future.

Reviewer: Furthermore, in response to the objection raised in the previous round, it is important to emphasize that in RPC, the separation of lipid species is primarily determined by the identity of the acyl chains and, to a lesser extent, by the sum composition. Therefore, a comparison based solely on the total number of carbon atoms cannot provide significant information about retention.

Reply: We agree wholeheartedly with this comment, and indeed this is the intention of our novel SVM-based workflow, which we developed precisely to combat this issue. We have rephrased the Discussion paragraph which sets out how the SVM based workflow operates and included the reviewer's text. We hope that - given our approach is in agreement with the reviewer's concerns - this will be satisfactory.

From Discussion, second paragraph:

"One approach to dealing with potential problems in LC-MS analysis is outlier detection. The novel SVM regression with LOOCV method described here successfully identified the major physicochemical properties governing elution order. This was achieved by using H count for acyl length, the CH relationship to identify saturation as a latent variable [49], and automatically identifying the hydrophobicity of lipid headgroups and their influence on tR, for example correctly ordering PC and DG headgroups. [50] Crucially, the algorithm can incorporate all these latent variables in its decision-making, instead of relying on one variable for assessment; a comparison based solely on C count for example cannot provide significant information about tR."